# Type I interferon regulates interleukin-1beta and IL-18 production and secretion in human macrophages

Rodrigo Díaz-Pino[1,2], Gillian I Rice[3,4], Diego San Felipe[1,2,5], Tamar Pepanashvili[1], Paul R Kasher[1,6,7], Tracy A Briggs[1,3,4], Gloria López-Castejón[1,2]

Inflammasomes are immune complexes whose activation leads to the release of pro-inflammatory cytokines IL-18 and IL-1β. Type I IFNs play a role in fighting infection and stimulate the expression of IFN-stimulated genes (ISGs) involved in inflammation. Despite the importance of these cytokines in inflammation, the regulation of inflammasomes by type I IFNs remains poorly understood. Here, we analysed RNA-sequencing data from patients with monogenic interferonopathies and found an up-regulation of several inflammasome-related genes. To investigate the effect of type I IFN on the inflammasome, we treated human monocyte-derived macrophages with IFN-α and observed an increase in *CASP1* and *GSDMD* mRNA levels over time, whereas *IL1B* and *NLRP3* were not directly correlated to IFN-α exposure time. IFN-α treatment reduced the release of mature IL-1β and IL-18, but not caspase-1, in response to ATP-mediated NLRP3 inflammasome activation, suggesting regulation occurs at cytokine expression levels and not the inflammasome itself. However, more studies are required to investigate how regulation by IFN-α occurs and impacts NLRP3 and other inflammasomes at both transcriptional and post-translational levels.

## Introduction

The innate immune system plays a fundamental role in recognising and responding to invading pathogens and damage signals. One important component of this response is the inflammasome, a multiprotein complex that upon activation and assembly generates a scaffold that facilitates the activation of caspase-1, leading to the maturation and secretion of pro-inflammatory cytokines IL-1β and IL-18. In addition, inflammasome activation leads to pyroptosis, a type of regulated lytic cell death, through the cleavage of gasdermin D (GSDMD) (Kayagaki et al, 2015). The NLRP3 inflammasome activation requires two signals: priming and activation. Priming is initiated by the binding of various ligands to their respective pattern recognition receptors, leading to signalling cascades that trigger gene transcription and increase the expression of inflammasome components and their post-translational modifications (Takeuchi & Akira, 2010; Shim & Lee, 2018). Bacterial LPS is widely used in research as it activates TLR4 leading to NFκB activation and up-regulation of different inflammasome components (Herman & Pasinetti, 2018). The second signal induces pro-translational changes that lead to inflammasome assembly. Among the multiple activators of the NLRP3 inflammasome are pathogen-associated molecular patterns derived from influenza virus, *Streptomyces hygroscopicus* (nigericin toxin), and Streptococcus *aureus* (Broz & Dixit, 2016), and damage-associated molecular patterns, such as ATP, silica crystals, uric acid crystals, aluminium hydroxide, amyloid-β, and bee venom (Mathur et al, 2018).

Type I IFNs are important cytokines known for their active role in regulating the immune response and key in the protection from pathogens, as they present a very strong antiviral activity. All the different type I IFNs, including IFN-α and IFN-β, signal through the type I IFN receptor (IFNAR). IFNAR activation leads to the formation of the IFN-stimulated gene factor 3 complex (ISG3), formed by STAT1/STAT2 and IRF9, that mediates the expression of a myriad of ISGs such as chemokines CXCL9 and CXCL10 or ISG15 (Lee & Ashkar, 2018).

Uncontrolled production of IL-1β and type I IFNs is responsible for the onset of autoinflammatory and autoimmune diseases. Autoinflammatory monogenic disorders characterised by high levels of IL-1β are well recognised and are termed "inflammasomopathies," such as cryopyrin-associated periodic syndromes consisting of gain-of-function mutations of NLRP3 (Voet et al, 2019; Alehashemi & Goldbach-Mansky, 2020; Krainer et al, 2020).

[1]Lydia Becker Institute of Immunology and Inflammation, Faculty of Biology, Medicine and Health, Manchester Academic Health Science Centre, The University of Manchester, Manchester, UK [2]School of Biological Sciences, Division of Infection, Immunity and Respiratory Medicine, Faculty of Biology, Medicine and Health, The University of Manchester, Manchester, UK [3]Department of Genomic Medicine, St Marys Hospital, Manchester Foundation Trust, Manchester, UK [4]Division of Evolution, Infection and Genomics, School of Biological Sciences, Faculty of Biology, Medicine and Health, The University of Manchester, Manchester, UK [5]Department of Physiology, Faculty of Medicine, Universidad Complutense de Madrid, Madrid, Spain [6]Geoffrey Jefferson Brain Research Centre, The Manchester Academic Health Science Centre, Northern Care Alliance and The University of Manchester, Manchester, UK [7]Division of Neuroscience, School of Biological Sciences, Faculty of Biology, Medicine and Health, The University of Manchester, Manchester, UK

Correspondence: gloria.lopez-castejon@manchester.ac.uk

Mutations in several genes of the IFN signalling pathway lead to a spectrum of diseases known as "type I interferonopathies," which result in unregulated production of type I IFN and ISGs (Crow & Stetson, 2022) and which are characterised by multiple symptoms of systemic inflammation (Raftopoulou et al, 2022). Among the syndromes recognised as type I interferonopathies are Aicardi–Goutières syndrome, spondyloenchondrodysplasia, or STING-associated vasculopathy.

Although L-1β and type I IFNs play divergent roles in the immune response, their cross-regulation is key to maintain an appropriate inflammatory response. For instance, it is well known that type I IFN can suppress pro-IL-1β expression, as well as induce IL-1Ra and IL-10 expression (Mayer-Barber & Yan, 2017). There is also evidence of a correlation between ISG expression, IL-1β and IL-18 production, and the inflammasome in patients with autoimmune disorders associated with elevated levels of IFNs such as systemic lupus erythematosus (SLE) and juvenile dermatomyositis (da Cruz et al, 2020; Verweyen et al, 2020; Roberson et al, 2022). However, and despite the importance of the inflammasome in IL-18 and IL-1β secretion, the regulation of the inflammasomes, and in particular NLRP3, by type I IFN is still unclear and contradictory. Murine type I IFN has been shown to not only inhibit the production of pro-IL-1β but also reduce caspase-1 activation by regulating NLRP3 at the translational level through STAT1 (Guarda et al, 2011). However, other studies suggest that IFN-α treatment enhances NLRP3 inflammasome activity, as evidenced by increased IL-1β release and caspase-1 cleavage in human monocytes and in epithelial cells in response to NLRP3 activators (Liu et al, 2017; Hong et al, 2020). Therefore, further understanding of how the NLRP3 inflammasome is regulated by type I IFN could provide new insights that may explain some of the inflammatory components of interferonopathies and could lead to novel treatments for patients.

In this study, we characterised the expression of inflammasome-related genes (IRGs) in whole blood RNA from patients with monogenic type I interferonopathies. We further characterised the effect of type I IFN on the expression of NLRP3 IRGs, as well as the activation of the NLRP3 inflammasome in primary monocyte-derived macrophages (MDMs) in vitro.

# Results

## Expression of IRGs is increased in whole blood obtained from patients with type I interferonopathies

As type I IFN has been shown to up-regulate inflammasome activation in human monocytes in vitro from healthy donors and in SLE patients characterised by high levels of type I IFN (Liu et al, 2017), we set to analyse the expression of inflammasome genes in patients with monogenic type I interferonopathies. For this, we used the RNAseq data set from our previously published project (Duncan et al, 2019). This was obtained from whole blood samples from age-matched healthy donors (n = 5) and patients with mutations in different genes that cause monogenic type I interferonopathies (ACP5, ADAR1, DNASE2, IFIH1, RNASEH2A, RNASEH2B,

RNASEH2C, RNASET2, SAMHD1, STAT2, TMEM173, TREX1, and PEPD [n = 42 overall]). The expression of the following ISGs was analysed: STAT1, STAT2, STAT3, IRF9, PKR, CGAS, STING, OAS1, OAS2, OAS3, IFI16, RSAD2, CXCL9, CXCL10, and ISG15. We observed an increased expression of ISGs in patients compared with healthy controls (Fig 1A and B), confirming the previously described type I IFN signature in these patients. In addition, PKR, RSAD2, CGAS, and STING genes were found to be up-regulated in patients compared with the control group (Figs 1A and S1). The analysis of inflammasome gene expression showed an up-regulation of the expression of inflammasome-associated genes (Fig 1C and D) NLRP3 (P = 0.0309), ASC (P = 0.0403), CASP1 (P = 0.0376), IL18 (P = 0.0458), and GSDMD (P = 0.0008) in donors with interferonopathies compared with healthy controls. In addition, other IRGs were analysed and there was a general trend towards increased expression compared with controls (Fig S1). Thus, our data suggest an association of NLRP3 inflammasome gene transcription with type I IFN in these patients.

## Type I IFN induces the gene expression of CASP1 and GSDMD in human macrophages in vitro

After the observation of an increased expression of inflammasome component genes in patients with monogenic interferonopathies, we undertook an in vitro assay to determine the direct effect of type I IFN on the expression of inflammasome genes in healthy macrophages eliminating other confounding factors present in these patients. We treated human blood MDMs from healthy donors, in the monocytic human cell line THP-1 (PMA-differentiated) frequently used as a model of human macrophages for 4 h with different concentrations of IFN-α and IFN-β, and then performed a quantitative polymerase chain reaction (qRT-PCR) to determine expression levels of ISGs and IRGs. To determine whether the results obtained were only applicable to macrophages, we also investigated gene expression in CD14+ human blood monocytes from healthy donors. We found a dose-dependent induced expression of the ISGs IRF9, ISG15, CXCL9, CXCL10, and USP18 in all cell types treated with type I IFN compared with untreated cells (UT) (Figs 2A and S2A and C). This shows that these cells are responsive to type I IFN.

Next, we analysed the expression of IRGs NLRP3, CASP1, GSDMD, IL18, and IL1B. We detected an up-regulation of CASP1 expression in MDMs, THP-1 cells, and human monocytes treated with high concentrations of IFN-α and IFN-β (100 and 1,000 U/ml) (Figs 2B and S2B and D). In addition, in MDMs we observed a trend towards an increased expression of GSDMD, which was significant in IFN-treated THP-1 cells (at 100 U/ml) (Fig 2B and D). NLRP3 expression decreased in MDMs when they were treated with IFN-β at high concentrations, but no effect on THP-1 cells or human monocytes was observed. In addition, IL18 expression tended to be reduced after treatment with type I IFN in both cell types. No effects on IL1B expression were detected in any of the cell types after IFN treatment. These data show that the gene expression of only CASP1 and GSDMD is up-regulated by type I IFN after short-term treatment of 4 h, and that NLRP3 expression is down-regulated by IFN-α in MDMs.

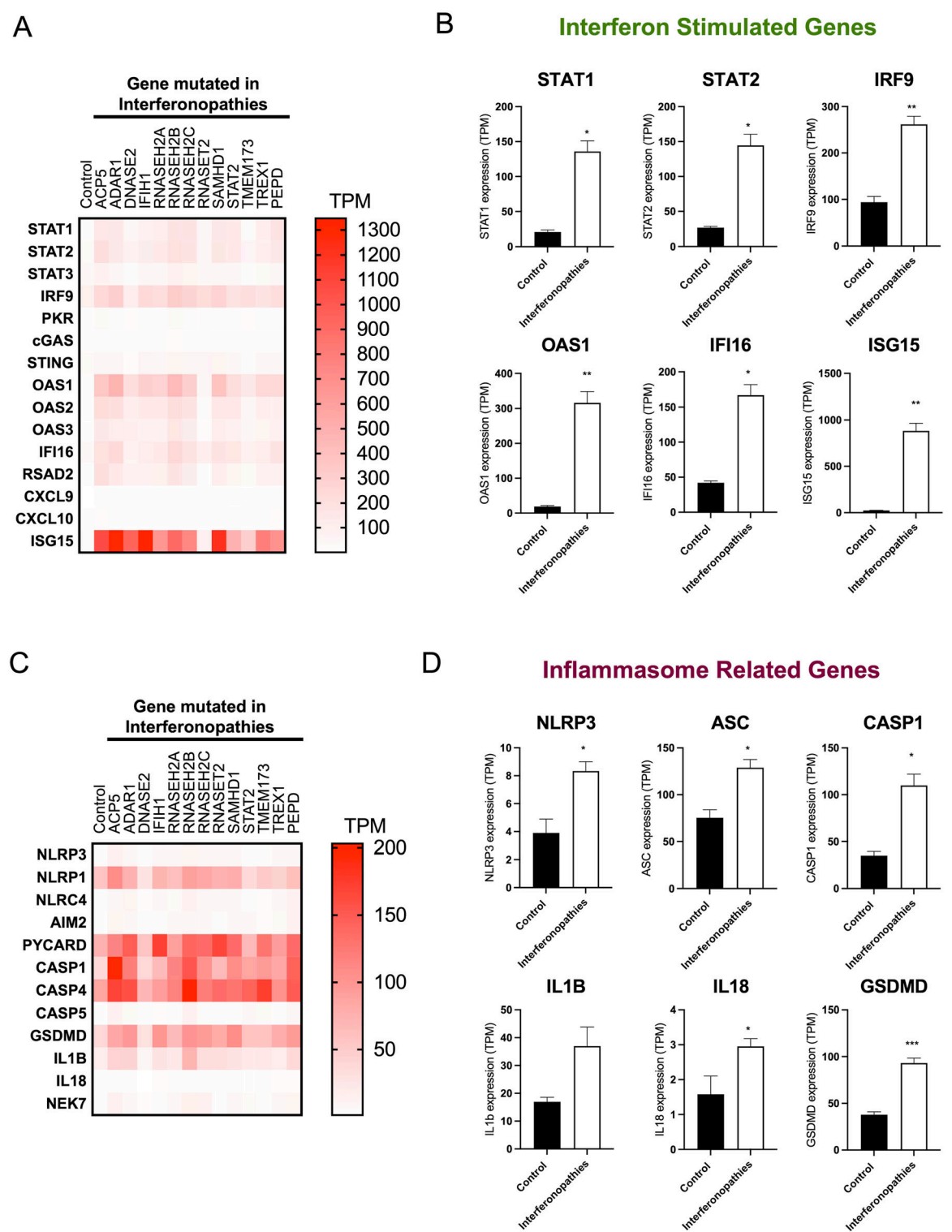

**Figure 1. Increased expression of inflammasome-related genes (IRGs) in patients with type I interferonopathies.**
RNAseq was performed from whole blood samples from patients with mutations in different genes that cause type I interferonopathies, *ACP5* (n = 3), *ADAR1* (n = 6), *DNASE2* (n = 3), *IFIH1* (n = 2), *RNASEH2A* (n = 3), *RNASEH2B* (n = 8), *RNASEH2C* (n = 6), *RNASET2* (n = 1), *SAMHD1* (n = 8), *STAT2* (n = 4), *TMEM173* (n = 3), *TREX1* (n = 8), and *PEPD* (n = 3). **(A)** Heatmap representation of the expression of indicated IFN-stimulated genes as transcripts per million (TPM) analysed from patients with the indicated genetic mutations. **(B)** Expression of IFN-stimulated genes was compared between donors with type I interferonopathies and healthy donors and is shown as TPM. **(C)** Heatmap representation of the expression of different IRGs as TPM analysed from patients with the indicated genetic mutations. **(D)** Expression of IRGs was compared between patients with type I interferonopathies and healthy donors and is shown as TPM. Healthy donors, n = 5; patients with type I interferonopathies, n = 42. Mean ± SEM; *P < 0.05 and **P < 0.005, *t* test.

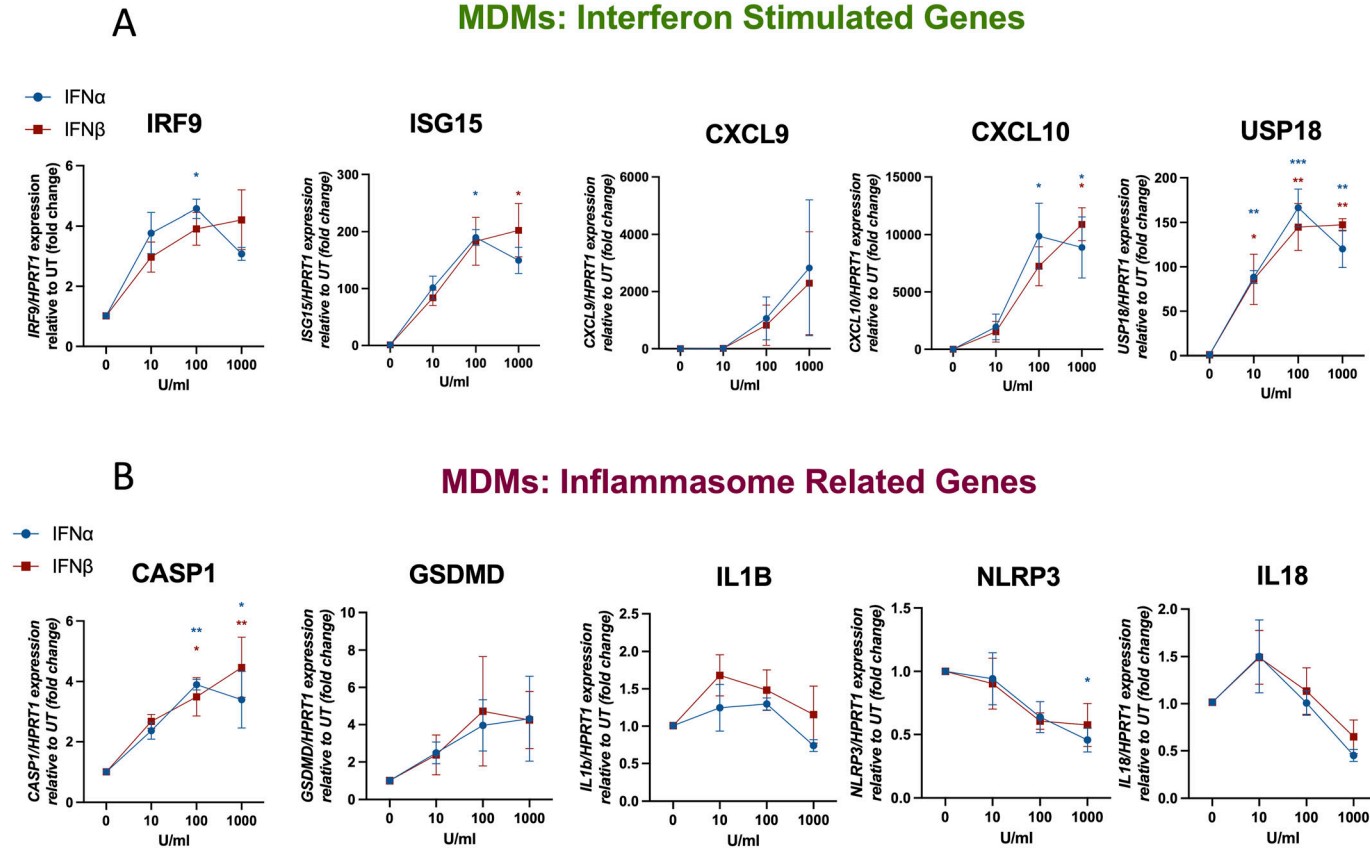

**Figure 2.  Type I IFN induces the expression of IFN-stimulated genes and inflammasome-related genes in human macrophages.**
**(A, B)** Human monocyte-derived macrophages from healthy donors (A, B) were treated for 4 h with either IFN-α (1,000, 100, or 10 U/ml) or IFN-β (1,000 or 100 or 10 U/ml) or left untreated (UT). Expression levels of mRNA for IFN-stimulated genes and inflammasome-related genes were measured by RT–qPCR and are shown as expression relative to UT (fold change). n = 3 biologically independent experiments. Mean ± SEM; *P < 0.05, **P < 0.005, and ***P < 0.0005, one-way ANOVA with Dunnett's post-test. HPRT1 was used as a housekeeping gene.

**LPS priming does not alter IFN-mediated up-regulation of *CASP1* and *GSDMD* gene expression in human macrophages**

After confirming that MDMs and PMA-differentiated THP-1 cells can respond to type I IFN and that this leads to an increase in *CASP1* and *GSDMD* gene expression, we decided to investigate whether this resulted in an altered activation of the NLRP3 inflammasome. As the priming step plays a key role in NLRP3 inflammasome activation, we first tested whether the observed effects of type I IFN on *CASP1* and *GSDMD* expression were altered by macrophage priming with LPS. To assess this, we used IFN-α because we did not see major differences in gene expression between IFN-α or IFN-β treatments (Fig 2) and it had been used in previous studies that showed inflammasome regulation by this cytokine (Liu et al, 2017). We treated human MDMs from healthy donors with IFN-α (1,000 U/ml) for 6 h (short term, ST) or overnight (ON, 18 h). MDMs were co-treated with IFN-α in the presence or absence of LPS (10 ng/ml) for the last 6 h of each different treatment (ST and ON) (Fig 3A). Samples were collected at this point, and qRT-PCR was performed to measure the expression of ISGs (Fig 3B) and IRGs (Fig 3C). We found that the different ISGs tested (*IRF9, ISG15, CXCL9, CXCL10, USP18,* and *STAT1*) were up-regulated after IFN-α treatment alone, as we found before

(Fig 2), and that this increase was time-dependent. The induced expression of ISGs by IFN-α alone was not altered by LPS treatment (Fig 3B). For the tested IRGs (*NLRP3, CASP1, GSDMD, IL1B, ASC,* and *IL18*), we found that *NLRP3* expression was significantly decreased in MDMs after treatment with IFN-α alone for 6 h (Fig 3C), similar to the effect observed at 4 h (Fig 2B), and then significantly increased when treated with IFN-α-ON compared with untreated cells and IFN-α-ST treatment (Fig 3C). LPS treatment for 6 and 4 h alone also led to a decrease in *NLRP3* mRNA that was reversed by IFN-α treatment overnight (Figs 3C and S3A). *CASP1* and *GSDMD* increased their expression with IFN-α treatment alone at both time points tested, and this was not altered by the presence of LPS. *IL1B* gene expression was reduced by IFN-α treatments alone at both time points. We observed that LPS alone induced an initial increase in *IL1B* gene expression at 4-h treatment (Fig S3B), followed by a reduction at 6-h treatment (Fig 3C) (although protein levels were still induced at this latest time point [Fig S3C]), and that these gene expression levels at 6 h were not changed by the treatment with IFN-α. IFN-α treatment did not alter *ASC* and *IL18* expression levels, even in the presence of LPS.

Having established the effect of LPS priming on the type I IFN–induced gene expression of NLRP3 inflammasome components, we

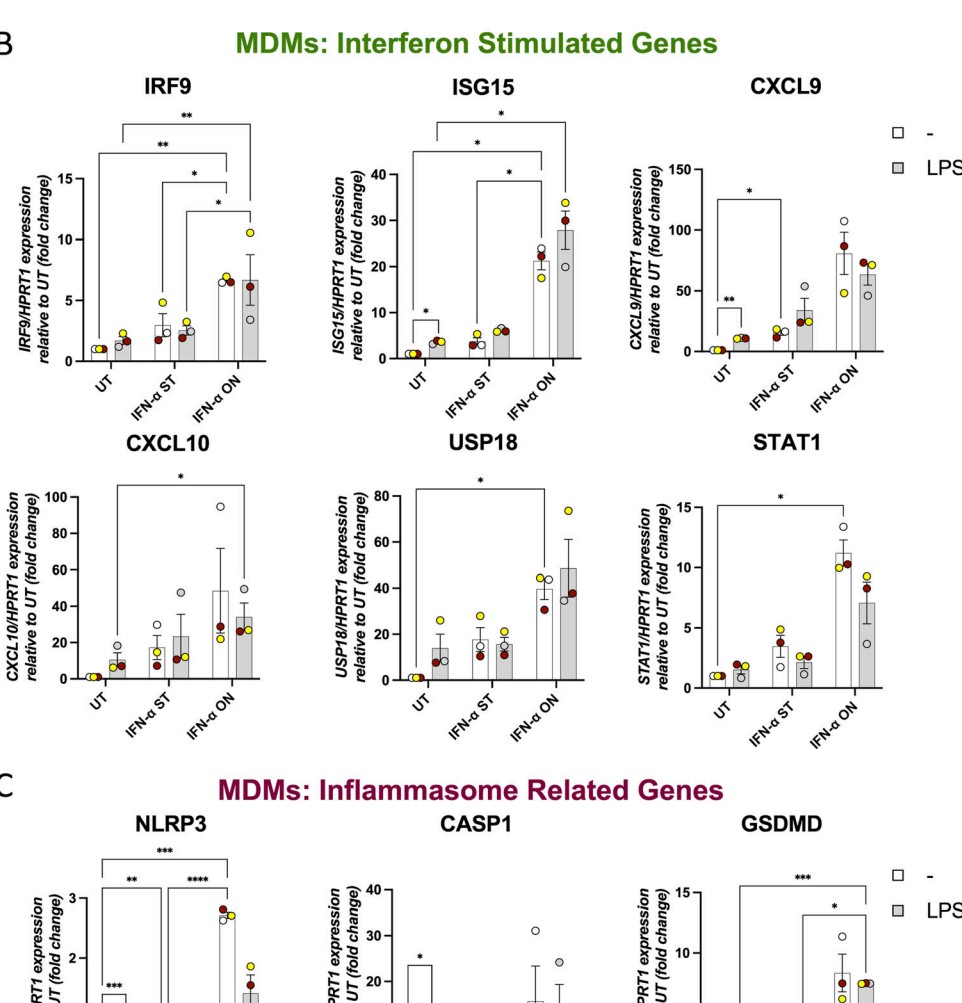

**Figure 3. LPS priming does not alter IFN-mediated up-regulation of caspase-1 and gasdermin D in human macrophages.** **(A)** Human monocyte-derived macrophages from healthy donors were left untreated or treated with IFN-α (1,000 U/ml) for 6 h (ST) or overnight (ON, 18 h) and with LPS (10 ng/ml) during the last 6 h of IFN treatment, before measuring gene expression (A). **(B, C)** Expression levels of mRNA for IFN-stimulated genes (B) and inflammasome-related genes (C) were measured by RT–qPCR and are shown as expression relative to UT (fold change). n = 3 biologically independent experiments. Mean ± SEM; *P < 0.05, **P < 0.005, and *** < 0.0005, two-way ANOVA with Tukey's post-test. *HPRT1* was used as a housekeeping gene.

next investigated whether the priming step alone, in the presence of type I IFN, led to inflammasome activation. To assess this, we treated human MDMs and CD14⁺ monocytes from healthy donors with IFN-α (1,000 U/ml) for 6 h (ST) or overnight (ON, 18 h). MDMs and monocytes were co-treated with IFN-α and LPS (10 ng/ml) or left untreated for the last 6 h of IFN treatments as in Fig 3A. Then, supernatants were collected to determine lactate dehydrogenase (LDH) enzyme release as a measure of cell death, caspase-1 activity, and the release of the pro-inflammatory cytokines IL-1β and IL-18 (Figs 4A–D and S4A–D). In MDMs, we found that none of the treatments induced cell death (Fig 4A), whereas we observed an increase in caspase-1 activity in cell supernatants (Fig 4B), as well as IL-1β release in MDMs treated with IFN-α-ON and LPS compared with the group treated with LPS alone (Fig 4C). In contrast, IL-18 release was not affected (Fig 4D). In human monocytes, no changes were observed in cell death or the release of IL-1β and IL-18, whereas caspase-1 activity was decreased when cells were pre-treated with IFN-α-ON compared with untreated monocytes or those treated with IFN-α-ST (Fig S4B).

### Type I IFN treatment alters IL-1β and IL-18 secretion after the activation of the NLRP3 inflammasome in human macrophages

We next wanted to see whether the regulation of the expression of the IRGs was reflected by an altered inflammasome activation response. Human MDMs and monocytes from healthy donors were treated with IFN-α-ON or IFN-α-ST and then in the last 6 h co-treated with LPS (10 ng/ml) to induce priming of the inflammasome (as in Fig 3), or left untreated, and then activated with ATP 5 mM for 2 h in the presence of LPS (Figs 4E and S4E). ATP is a known ligand of the P2X7 receptor and leads to the activation of the inflammasome in a less saturating way than the nigericin toxin (Herman & Pasinetti, 2018), which is useful to monitor the effects of type I IFN without being masked by a rapid cell death that could be caused by a more potent activator. In MDMs, treatment of LPS-primed cells with ATP was successful in activating the inflammasome as it increased cell death, caspase-1 activity, and the release of the cytokines IL-1β and IL-18 (Fig 4F–I) when compared to the control. This was also confirmed by Western blot of IL-1β (Fig 4J and K) in supernatants. We observed that IFN treatment led to a slight decrease in LDH release after ATP activation (Fig 4F), as well as a reduction in secretion of IL-1β and IL-18 in response to ATP (Fig 4H and I). IFN treatment, however, did not alter caspase-1 activity induced by ATP, but increased caspase-1 activity induced by LPS alone (Fig 4G). Meanwhile, in human monocytes, a decrease in cell death, as well as a decrease in caspase-1 activity, was observed in monocytes treated with IFN-α-ON in the absence of LPS priming (Fig S4F and G). Despite this, no changes were observed in the release of IL-1β (Fig S4H) and IL-18 (Fig S4I) triggered by LPS and ATP.

To determine whether the absence of regulation of inflammasome activation by IFN-α was just limited to NLRP3 activation by P2X7R, we next tested the effect of IFN treatment on inflammasome activation by the more potent NLRP3 trigger, nigericin (Herman & Pasinetti, 2018). Here, MDMs from healthy donors were treated for 4 h or ON with IFN-α (1,000 U/ml) and LPS was added for the last 2 h of the treatment followed by nigericin treatment (10 μM) for 45 min (Fig 5A). We observed that the inflammasome was activated in

response to LPS priming and nigericin as expected and as evidenced by increased cell death, caspase-1 activity, and the release of IL-1β and IL-18 (Fig 5B–I). Pre-treatment with IFN-α at different time points had no effect on the levels of inflammasome activation mediated by nigericin (Fig 5B–I). These experiments were also carried out in PMA-differentiated THP-1 cells (Fig S5A–I). Although the inflammasome was activated after LPS and nigericin treatment, no type I IFN–mediated changes were observed in this cell type either.

### Differentiation of CD14⁺ cells into macrophages in the presence of type I IFN does not affect MDM response to inflammasome activation

Interferonopathies are characterised by the presence of persistently high levels of IFN in the body (d'Angelo et al, 2021; Crow & Stetson, 2022). Hence, we wanted to determine the effect that differentiation of human CD14⁺ monocytes into MDMs in the presence of IFN has on the subsequent inflammasome response. Thus, human monocytes from healthy donors were treated with M-CSF (0.5 ng/ml) as usual (Sierra-Filardi et al, 2014) in the presence or absence of IFN-α at 1 and 5 U/ml. After 3 d, half of the medium was removed and supplemented again with M-CSF and type I IFN as before. After 7 d, no change in cell death (Fig 6A and B) or basal IL-1β (Fig 6C) or IL-18 (Fig 6D) release was observed compared with MDMs differentiated in the absence of IFN-α. The expression of ISGs IRF9, ISG15, CXCL9, CXCL10, and USP18 was analysed via qRT-PCR to confirm the effect of IFN-α treatment on these cells. A significant increase in the expression of IRF9 and ISG15 was observed after treatment with 1 or 5 U/ml of IFN-α, whereas CXCL9, CXCL10, and USP18 only showed a tendency to increase, but this was not significant (Fig 6E). Subsequently, MDMs were treated with LPS (10 ng/ml) for 2 h to induce priming followed by treatment with nigericin (10 μM) for 45 min to activate the inflammasome. We observed the expected effects of nigericin on inflammasome activation through increased cell death, caspase-1 activity, and the release of IL-1β and IL-18 (Fig 6F and G) in all different treatments; however, there were no differences in the levels of inflammasome activation between cells differentiated in the presence of IFN-α at 1 U/ml (Fig 6F), IFN-α at 5 U/ml (Fig 6G), or those differentiated in the absence of type I IFN. This suggests that in vitro differentiation of monocytes into MDMs in the presence of type I IFN does not alter NLRP3 inflammasome activation in this experimental context.

## Discussion

Although the crosstalk between type I IFN and inflammasomes in response to infection is somewhat established (Mazewski et al, 2020), little is known about how type I IFN and the NLRP3 inflammasome interact in the context of sterile inflammation and autoinflammatory disease. Here, we show that type I IFN can induce changes to inflammasome gene expression and reduce NLRP3-mediated IL-1β and IL-18 release.

The role of type I IFN in NLRP3 inflammasome regulation remains controversial. Initially, it was proposed that type I IFN down-

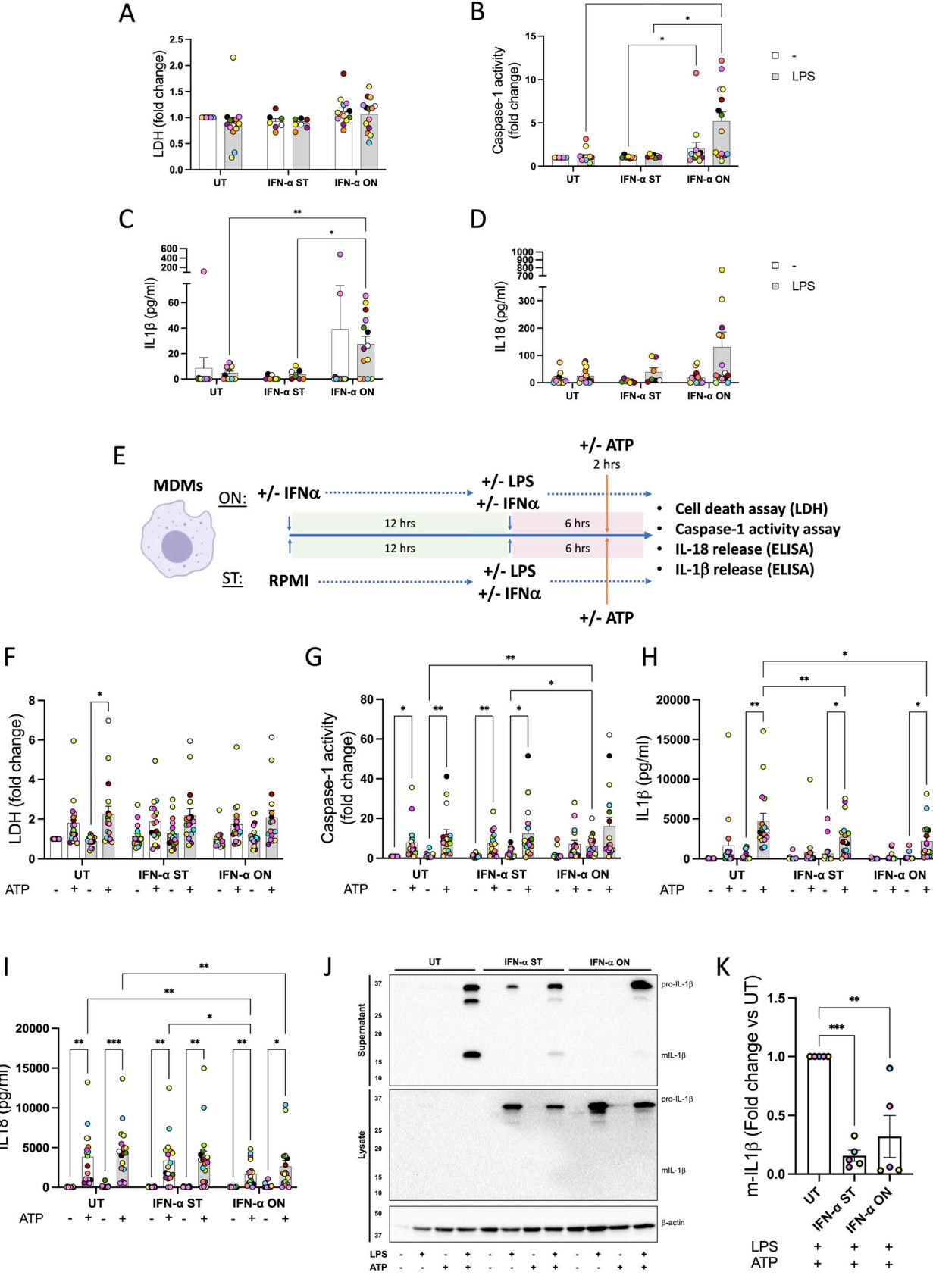

regulates NLRP3 inflammasome activation. The initial discovery showed that IFN-α and IFN-β treatment impaired NLRP3 inflammasome assembly in BMDMs via STAT1 (Guarda et al, 2011). This was followed by a report showing that the expression of *NLRP3*, *CASP1*, and *IL1B* in PBMCs from SLE patients was reduced compared with healthy donors (Yang et al, 2014), which is interesting given a proportion of SLE patients are known to have high type I IFN levels (Dall'Era et al, 2005). Later reports, however, presented different conclusions. Liu et al showed that treatment of healthy human monocytes with IFN-α potentiated IL-1β release and caspase-1 activation. This study also showed that monocytes from SLE patients had a higher expression of IRGs and higher inflammasome activation than healthy donors, and that this response was further potentiated by IFN-α treatment (Liu et al, 2017). Similar results were found by Hong et al, which showed that type I IFN treatment leads to an increased expression of *CASP1* and *GSDMD*, in human salivary gland epithelial cells, and that this correlates with enhanced NLRP3 inflammasome activation in these cells (Hong et al, 2020).

Our results are in accordance with these latest studies as we found that patients with monogenic type I interferonopathies exhibit an increased expression of IRGs compared with healthy donors and that IFN-α or IFN-β treatment leads to an increase in mRNA expression levels of *CASP1* and *GSDMD* in healthy MDMs. Interestingly, we found that IFN-α treatment did not regulate all IRGs in the same way. In MDMs, IFN-α treatment led to an initial decrease in *NLRP3* mRNA expression (4–6 h) and to a later increase in expression compared with untreated cells, similar to what happened to *NLRP3* expression after LPS treatment alone. Likewise, *IL1B* mRNA expression decreased after treatment with IFN-α alone. LPS treatment at 4 h induced an initial increase in *IL1B* transcript, which decreased after 6 h alone. This is in alignment with previous research that showed *IL1B* mRNA half-life is ~4 h, and despite this decrease in mRNA levels, pro-IL1B levels are still sustained after this decay (Schindlers et al, 1989; Hadadi et al, 2016). In addition, different LPS concentrations can lead to different patterns of transcriptional changes in human monocytes (Naler et al, 2022). It is then possible that *NLRP3* and *IL1B* follow different patterns of expression after the low dose of LPS used here (10 ng/ml), compared with what is usually used (1,000 ng/ml) in inflammasome research (Song et al, 2017); however, this was not reflected at protein levels in our hands (Fig S3C). This reflects the complexity of gene expression and protein-level correlation in general (Liu et al, 2016), and more specifically of different inflammatory genes and

how regulation of such genes by type I IFN requires further studies. Of note, we found that long-term IFN-α treatment followed by LPS, in the absence of an activating signal, did induce IL-1β and caspase-1 release. It is known that LPS treatment alone is able to induce inflammasome activation in human monocytes (Yu et al, 2021), and to a lesser extent in human macrophages (Schroder et al, 2012). It is hence possible that IFN could potentiate inflammasome activation in these conditions. However, we still do not know whether this effect is induced by the activation of NLRP3 as we cannot exclude the implication of other inflammasomes such as AIM2 or the involvement of non-canonical inflammasomes (Cui et al, 2014). Moreover, how the changes between mRNA, protein expression, and inflammasome assembly after different LPS and IFN treatments are correlated is still unknown.

We observed that IFN-α treatment led to a decrease in IL-1β and IL-18 levels after NLRP3 inflammasome activation by ATP. However, this pattern was not followed by caspase-1 activity present in the supernatants. This suggests that the decrease in cytokine secretion could be due to the down-regulation of the pro-IL-1β and pro-IL18 levels, and not the regulation of the NLRP3 inflammasome. We know the regulation of NLRP3 components at the transcriptional level (priming) is less relevant in human than in murine cells (Gritsenko et al, 2020); it is then possible that the observed changes in the gene expression of the inflammasome have little impact on final inflammasome assembly. In addition, we cannot discard the involvement of other inflammasomes such as AIM2 in this process. We also tested the effect of nigericin in this context. We found that for MDMs, no significant changes in IL-1β and IL-18 release were observed. This could be due to this experiment being underpowered and the high variability encountered among human samples; however, similar experiments in THP-1 cells, which we know are far less variable, also showed no effect in response to IFN-α (and IFN-β treatment, not shown), suggesting these effects might be specific to the P2X7R-mediated activation of the inflammasome in macrophages.

When we tested the effect of type I IFN on CD14[+] human monocytes, we observed a decrease in extracellular caspase-1 activity after type I IFN-α-ON treatment, but this was not reflected on IL-1β and IL-18 release levels. Macrophages and monocytes can present different behaviours regarding inflammasome activation and contribution to inflammation, which could explain the differences observed between MDMs and monocytes. For instance, MDMs release lower IL-18 levels compared with

---

**Figure 4. Type I IFN treatment alters IL-1 release induced by the ATP-mediated NLRP3 inflammasome in macrophages.**
Human monocyte-derived macrophages, monocyte-derived macrophages from healthy donors, were treated with IFN-α (1,000 U/ml) for 6 h (ST) or overnight (ON, 18 h) and with LPS (10 ng/ml) during the last 6 h of treatment with type I IFN to induce inflammasome priming. **(A)** Cell death was measured by the lactate dehydrogenase assay and is shown as a fold change relative to UT. **(B)** Caspase-1 release was measured using a quantitative assay and is shown as a fold change relative to UT. **(C)** IL-1β release was measured by ELISA in the supernatants and is shown as pg/ml. **(D)** IL-18 release was measured by ELISA in the supernatants and is shown as pg/ml. n = 14 biologically independent experiments. **(E)** Human monocyte-derived macrophages from healthy donors were treated with IFN-α (1,000 U/ml) for 6 h (ST) or overnight (ON, 18 h), and during the last 6 h of treatment with type I IFN, LPS (10 ng/ml) was added to induce inflammasome priming, and ATP (5 mM) was added in the last 2 h of the experiment. **(F)** Cell death was measured by the lactate dehydrogenase assay and is shown as a fold change relative to UT. **(G)** Caspase-1 activity in the supernatants was measured using a quantitative assay and is expressed as a fold change relative to UT. **(H, I, J)** Release of IL-1β and (I) IL-18 was measured by ELISA. n = 17 biologically independent experiments. (J) IL-1β processing was measured in supernatants and cell lysates by Western blot (WB). Representative of the four different biologically independent experiments. Mean ± SEM; *P < 0.05, **P < 0.005, and ***P < 0.0005, two-way ANOVA with Tukey's post-test. No comparison between unprimed and LPS-primed groups has been shown. **(K)** Densitometry of the mature form of IL-1β in the supernatants (mIL-1β) in the UT, IFN-α-ST, and IFN-α-ON groups treated with LPS and ATP and is shown as a fold change relative to UT. Mean ± SEM; *P < 0.05, **P < 0.005, and ***P < 0.0005, one-way ANOVA with Dunnett's post-test.
Source data are available for this figure.

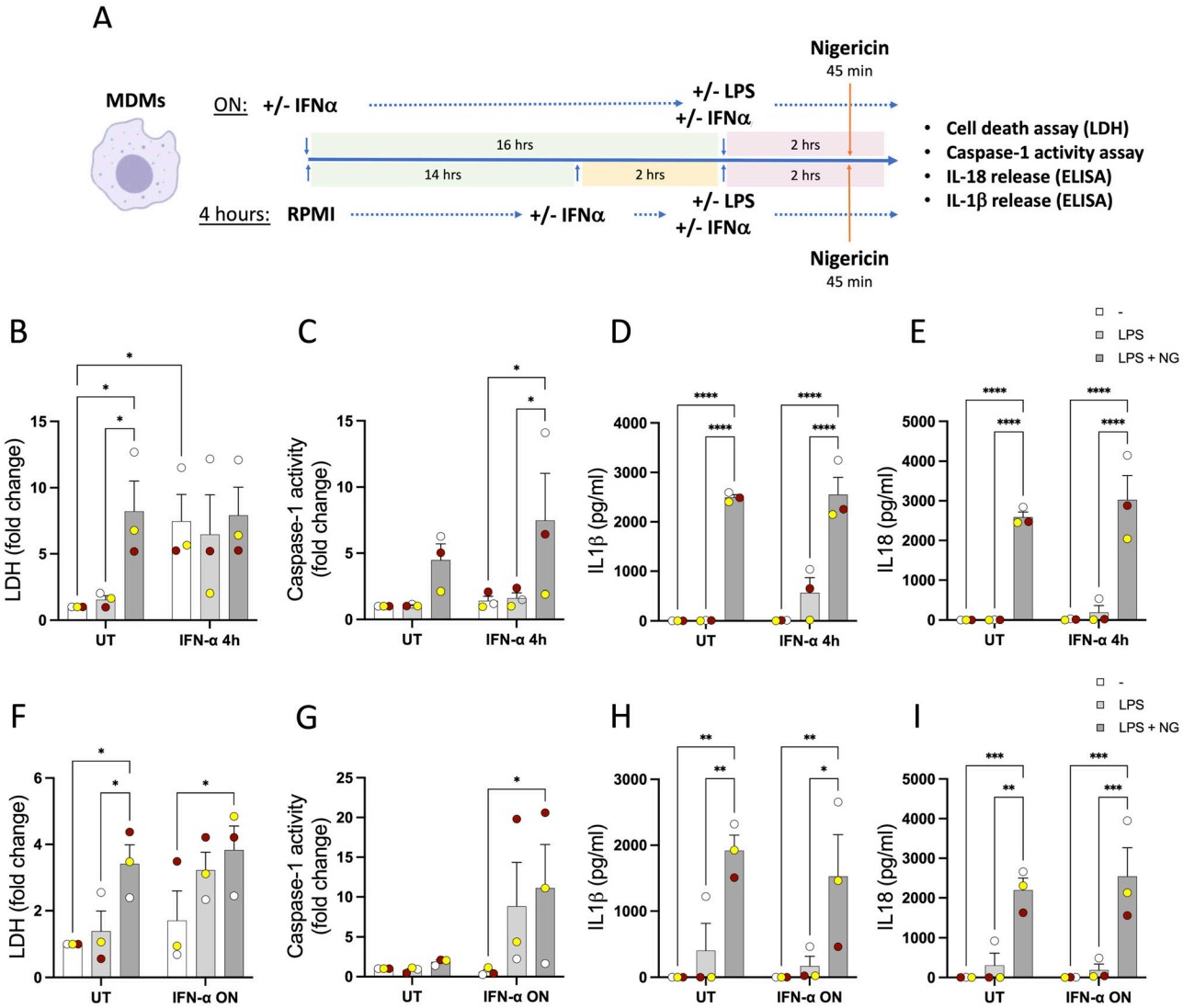

**Figure 5.  Type I IFN treatment does not alter the activation of the NLRP3 inflammasome triggered by nigericin in human macrophages.**
**(A)** Monocyte-derived macrophages derived from healthy donors were treated for 4 or 18 h (ON) with IFN-α (1,000 U/ml), then LPS (10 ng/ml) was used in conjunction with type I IFN for 2 h, and finally, nigericin was used for 45 min to activate the NLRP3 inflammasome. **(B, F)** Cell death was measured by the lactate dehydrogenase assay and is shown as a fold change relative to UT. **(C, D, E, G, H, I)** Caspase-1 activity in the supernatants was measured and is shown as a fold change relative to UT. The release of IL-1β (D, H) and IL-18 (E, I) was measured by ELISA. n = 3 biologically independent experiments. Mean ± SEM; *P < 0.05, **P < 0.005, and ***P < 0.0005, two-way ANOVA with Tukey's post-test.

undifferentiated monocytes in response to ATP activation (Awad et al, 2017).

As cells from SLE and monogenic interferonopathy patients are chronically exposed to type I IFN (Dall'Era et al, 2005), we investigated whether longer exposures to type I IFN could influence further responses to inflammasome activation. Initially, we differentiated CD14⁺ monocytes in high concentrations of IFN-α (100 U/ml); however, under these conditions IFN treatment induced cell death and MDMs were not viable by the end of the differentiation process (data not shown). When we reduced IFN-α concentrations to 1 and 5 U/ml, we did not observe cell death after 7 d. However, no significant differences between untreated or IFN-differentiated cells were observed when conducting nigericin-induced NLRP3

inflammasome activation, indicating that although type I IFN treatment can alter the transcriptional profile of macrophages in the presence of M-CSF (Fig 6, Tong et al, 2019), it was not sufficient to induce changes in nigericin-mediated NLRP3 inflammasome activation. Whether this would be sufficient to alter responses to the ATP-mediated NLRP3 inflammasome would need to be further investigated.

In conclusion, our study shows that type I IFN cross-talks with inflammasome components via gene regulation, that IFN dampens the NLRP3-mediated release of IL-1β and IL-18 in response to ATP, and that IFN can potentiate Il-1 release in response to less acute inflammatory signals such as TLR activation alone. Our study highlights the need for further studies to better understand the

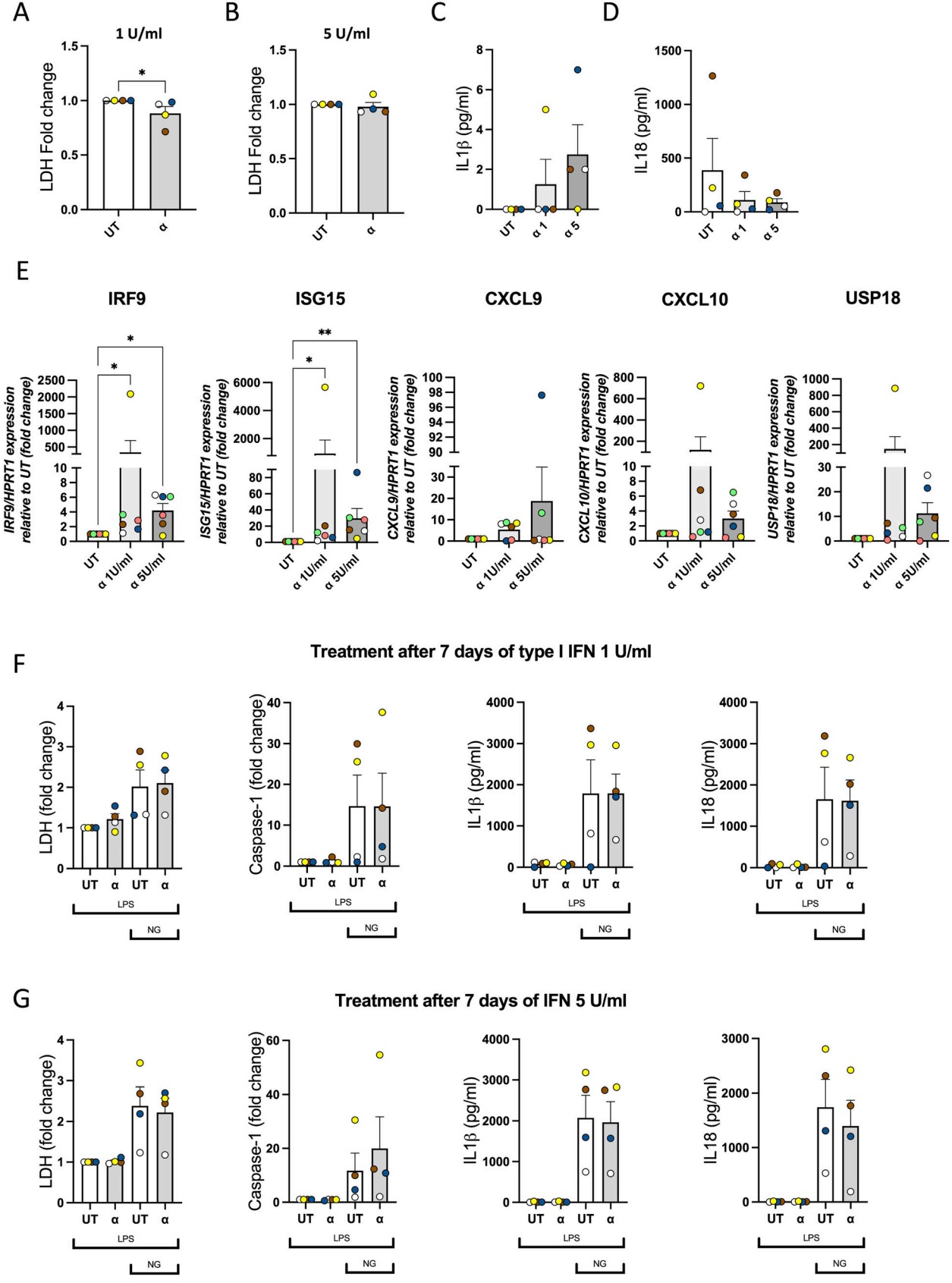

relationship between type I IFN and the NLRP3 inflammasome at transcriptional, post-transcriptional, and post-translational levels, as well as the involvement of other inflammasomes such as AIM2. For this, it is important to consider relevant differences between the type of cells investigated, the nature of the inflammasome activators, and the differences between human and murine systems where most of the NLRP3 activation dogmas have been established. This will allow us to gain a better understanding of how such important regulators of inflammation collaborate to maintain an appropriate inflammatory response.

## Materials and Methods

### Reagents and antibodies

LPS (*Escherichia coli* 026:B6), adenosine 5′-triphosphate disodium salt hydrate (A2383; ATP), nigericin (N7143), and penicillin–streptomycin (Pen/Strep, P4333) were purchased from Sigma-Aldrich. Human IFN Beta (Beta 1a, Mammalian Expressed) (11415-1) and Human IFN Alpha Hybrid (Universal Type I IFN) (11200-1) were obtained from PBL Assay Science. Phosphatase inhibitor cocktail (78420) was sourced from Thermo Fisher Scientific, and FBS was obtained from Gibco.

Primary antibodies for Western blot analysis and their final concentrations were as follows: anti-human IL-1$\beta$ (0.1 $\mu$g/ml, goat polyclonal, AF-201-NA; R&D Systems) and anti-$\beta$-actin HRP (0.2 $\mu$g/ml, mouse monoclonal, A3854; Sigma-Aldrich).

### Cell cultures and treatments

THP-1 cells were purchased from ATCC and were cultured in RPMI with 10% FBS and Pen/Strep (100 U/ml) and plated at a concentration of $1 \times 10^6$ cells/ml. PBMCs were obtained from the National Blood Transfusion Service with full ethical approval from the Proportionate Review Sub-committee of the London—South East Research Ethics Committee (REC reference: 21/PR/0526). PBMCs were isolated from leucocyte cones from healthy donors by density centrifugation using a Ficoll gradient. The PBMC layer was separated and washed to remove platelets. Monocytes were harvested by incubation of the PBMCs with magnetic CD14 MicroBeads (130-050-201; Miltenyi) for 15 min at 4°C. Monocytes were positively selected for using an LS column (130-042-401; Miltenyi). For the experiments where only monocytes were used, they were cultured (at a concentration of $1 \times 10^6$ cells per millilitre) in RPMI supplemented with 10% FBS, Pen/Strep (100 U/ml). To differentiate MDMs, monocytes were cultured for 6 d (at a concentration of $5 \times 10^5$ cells per millilitre) in RPMI supplemented with 10% FBS, Pen/Strep (100

U/ml) and treated with 0.5 ng/ml M-CSF (300-25; PeproTech). On day 3, half of the media was removed and replaced with fresh media.

Cells were untreated or incubated for 4 h with IFN-$\alpha$ (1,000 or 100 or 10 U/ml) or IFN-$\beta$ (1,000 or 100 or 10 U/ml). Total RNA was isolated and used for real-time qRT-PCR.

In ATP activation experiments, human monocytes or MDMs were untreated or pre-incubated with IFN-$\alpha$ (1,000 U/ml) for 6 h (short term, ST) or overnight (18 h) (ON). In the last 6 h of incubation, cells were left untreated or primed for 2 h with LPS (10 ng/ml) in ET buffer (147 mM NaCl, 10 mM Hepes, 13 mM D-glucose, 2 mM KCl, 2 mM CaCl$_2$, and 1 mM MgCl$_2$). Cells were treated with ATP (5 mM) in the last 2 h of the experiment.

In nigericin activation experiments, THP-1 cells and MDMs were untreated or pre-incubated with IFN-$\alpha$ (1,000 U/ml) for 4 h or overnight (18 h) (ON). In the last 2 h of incubation, cells were untreated or primed for 2 h with LPS (10 ng/ml). The priming stimulus was then removed and replaced with ET buffer. Cells were treated with nigericin (10 $\mu$M, 45 min).

### RNA isolation and real-time qPCR

Total RNA was isolated from THP-1 cells, human monocytes, and MDMs using RNeasy Mini Kit (74104; QIAGEN) according to the manufacturer's instructions. 500 ng of total RNA was reverse-transcribed into cDNA using High-Capacity RNA-to-cDNA Kit (4387406; Applied Biosystems). Real-time qPCR was performed in triplicate using Fast SYBR Green Master Mix (4385612; Thermo Fisher Scientific). The sequence of the primers used in this study is described in Table 1. Data were normalised to expression levels of the housekeeping gene HPRT1 (QT00059066) across each treatment (Ishii et al, 2006). Representative melt curve plots for each of the primers used can be found in Fig S6. Gene expression was calculated by the comparative threshold cycle (Ct) method, and fold change compared with basal RNA levels of untreated cells was expressed as $2^{-\Delta\Delta Ct}$.

### RNA sequencing

Whole blood samples from three patients with mutations in *ACP5*, six with mutations in *ADAR1*, three with mutations in *DNASE2*, two with mutations in *IFIH1*, three with mutations in *RNASEH2A*, eight with mutations in *RNASEH2B*, six with mutations in *RNASEH2C*, one with mutation in *RNASET2*, eight with mutations in *SAMHD1*, four with mutations in *STAT2*, three with mutations in *TMEM173*, eight with mutations in *TREX1*, three with mutations in *PEPD*, and five age-matched controls were analysed. RNA integrity was analysed with Agilent 2100 Bioanalyzer (Agilent Technologies). mRNA purification

**Figure 6. Differentiation of CD14$^+$ cells into macrophages does not affect monocyte-derived macrophages (MDMs) in response to inflammasome activation.**
**(A)** Human CD14$^+$ monocytes derived from healthy donors were differentiated into macrophages (MDMs) with M-CSF (0.5 ng/ml) in the presence of IFN-$\alpha$ at concentrations of 1 U/ml (A) and 5 U/ml. **(B)** Cell death was measured by the lactate dehydrogenase assay and is shown as a fold change relative to UT after 7 d of differentiation. **(C, D)** Release of IL-1$\beta$ (C) and IL-18 (D) was measured by ELISA. **(E)** Expression levels of mRNA for IFN-stimulated genes *IRF9*, *ISG15*, *CXCL9*, *CXCL10*, and *USP18* were measured by RT–qPCR and are shown as expression relative to UT (fold change). **(F, G)** Differentiated MDMs in the presence of type I IFN 1 U/ml (F) and 5 U/ml (G) were treated with LPS (10 ng/ml) for 2 h followed by treatment with nigericin (10 $\mu$M) for 45 min. Cell death was measured by the lactate dehydrogenase assay and is shown as a fold change relative to UT. The release of IL-1$\beta$ and IL-18 was measured by ELISA. n = 4 biologically independent experiments. Mean ± SEM; *$P$ < 0.05, **$P$ < 0.005, and ***$P$ < 0.0005, one-way ANOVA with the Kruskal–Wallis post-test.

Table 1. List of primers used in this study. Primer for qRT–PCR.

| Gene | Primer | Sequence (5′-3′) |
|---|---|---|
| IRF9 | Forward | GCCCTACAAGGTGTATCAGTTG |
| | Reverse | TGCTGTCGCTTTGATGGTACT |
| ISG15 | Forward | CACCGTGTTCATGAATCTGC |
| | Reverse | CTTTATTTCCGGCCCTTGAT |
| CXCL9 | Forward | CAGTAGTGAGAAAGGGTCGC |
| | Reverse | AGGGCTTGGGGCAAATTGT |
| CXCL10 | Forward | GTGGCATTCAAGGAGTACCTC |
| | Reverse | TGATGGCCTTCGATTCTGGATT |
| USP18 | Forward | TCCCCCAGAGCTTGGATTT |
| | Reverse | GCATCACAAGACTCTCGCTTCA |
| NLRP3 | Forward | TGCCCGTCTGGGTGAGA |
| | Reverse | CCGGTGCTCCTTGATGAGA |
| ASC | Forward | GCCAGGCCTGCACTTTATAGA |
| | Reverse | GTTTGTGACCCTCGCGATAAG |
| CASP1 | Forward | ATACCAAGAACTGCCCAAGTTTG |
| | Reverse | GGCAGGCCTGGATGATGA |
| IL1B | Forward | ACGATGCACCTGTACGATCACT |
| | Reverse | CACCAAGCTTTTTGCTGTGAGT |
| IL18 | Forward | AAGGAAATGAATCCTCCTGATAACA |
| | Reverse | CCTGGGACACTTCTCTGAAAGAA |
| GSDMD | Forward | ATGAGGTGCCTCCACAACTTCC |
| | Reverse | CCAGTTCCTTGGAGATGGTCTC |

and fragmentation, complementary DNA (cDNA) synthesis, and target amplification were performed using Illumina TruSeq RNA Sample Preparation Kit (Illumina). Pooled cDNA libraries were sequenced using the HiSeq 4000 Illumina platform (Illumina). For this, we used the RNAseq data set from our previously published project (Duncan et al, 2019). Written informed consent for these studies was provided, and ethical/institutional approval was granted by the South Central-Hampshire A (ref: 17/SC/0026) and Leeds (East) (ref: 07/Q1206/7).

## Cell death assay

The supernatant was centrifuged for 5 min at 500$g$ at 4°C to remove any cells. Cell death was quantified using a colorimetric assay for the release of LDH into cell supernatants using CytoTox 96 Non-radioactive Cytotoxicity Assay (G1780; Promega), according to the manufacturer's instructions. Absorbance values were measured at 490 nm, and the results were expressed as a percentage normalised to total cell lysis or as a fold change relative to untreated cells.

## Caspase-1 inflammasome assay

Caspase-1 release was measured using a quantitative assay of cell supernatants. Caspase-Glo 1 Inflammasome Assay (G9951; Promega) was used according to the manufacturer's instructions. The

luminescence of the assay plates was read. The results were expressed as a fold change relative to untreated cells.

## ELISA

Levels of human IL-1$\beta$ (DY201) and IL-18 (DY318) were measured in the cell supernatants using ELISA kits from R&D Systems. ELISAs were performed following the manufacturer's instructions.

## Western blot analysis

Cells were lysed on ice using a RIPA lysis buffer (50 mM Tris–HCl, pH 8, 150 mM NaCl, 1% NP-40, 0.5% sodium deoxycholate, and 0.1% SDS), supplemented with a protease inhibitor cocktail (P8340, 1:100; Sigma-Aldrich). Lysates were then centrifuged at 21,000$g$ for 10 min to remove the insoluble fraction. Protein concentrations of each sample were measured using the BCA assay (23225; Thermo Fisher Scientific Pierce), following the manufacturer's guidelines, to standardise the amount of protein in each sample. Cell supernatants were centrifuged at 500$g$ for 5 min to remove cells and concentrated using 10-kD MW cut-off filters (Amicon, Merck Millipore), as described by the manufacturer. Supernatants and lysates were diluted in 1× reducing Laemmli buffer containing 1% $\beta$-mercaptoethanol. Samples were heated at 95°C for 5 min and separated by Tris-glycine SDS–PAGE. Proteins were transferred onto nitrocellulose membranes (0.2 $\mu$m), followed by blocking with PBS–Tween (0.1%) containing 5% skimmed milk for 1 h at RT. Membranes were then incubated overnight with the specific primary antibody in blocking buffer at 4°C. The next day, membranes were labelled with an HRP-conjugated secondary antibody for 1 h at RT. After washing, membranes were imaged using Clarity Western ECL Blotting Substrate (1705061; Bio-Rad) in ChemiDoc MP Imager (Bio-Rad). Densitometry analysis was performed using ImageJ (rsb.info.nih.gov) to measure the intensity of mIL-1$\beta$ bands. Processed IL-1$\beta$ release was compared with control treatment and expressed as fold change.

## Statistical analysis

GraphPad Prism 10 software was used to carry out all statistical analysis. Data were first tested for normality using the Shapiro–Wilk test. Differences between 3+ groups were analysed using one-way ANOVA with the post hoc Dunnett test or two-way ANOVA with the post hoc Tukey test for multiple comparisons. Data were shown as the mean ± SEM. Two-way ANOVA used a multiple comparison. Accepted levels of significance were *$P < 0.05$, **$P < 0.01$, ***$P < 0.001$, and ****$P < 0.0001$. Where no significance levels are indicated, no significance difference was detected. In Figs 4 and S4, significance levels are only shown within the unprimed and LPS-primed treatments, but not between them. In addition, significance was indicated between the UT, IFN-$\alpha$-ST, and IFN-$\alpha$-ON groups.

# Supplementary Information

# Acknowledgements

This study was supported by Agencia Nacional de Investigación y Desarrollo de Chile (ANID) (Becas Chile 72200337) to R Díaz-Pino, a mobility grant (EB14/23) associated with his predoctoral contract for research personnel in training from the Complutense University of Madrid and Banco Santander (CT63/19-CT64/19) to D San Felipe, the MRC (MR/T03291X/1) award to PR Kasher, the MRC (MR/T016043/1) award to G López-Castejón, and the UK National Institute of Health Research TRF-2016-09-002, the NIHR Manchester Biomedical Resource Centre, and the Medical Research Foundation awards to TA Briggs.

## Author Contributions

R Díaz-Pino: formal analysis, investigation, and writing—original draft, review, and editing.
GI Rice: investigation.
D San Felipe: data curation and investigation.
T Pepanashvili: data curation and investigation.
PR Kasher: conceptualisation, resources, supervision, and writing—original draft, review, and editing.
TA Briggs: conceptualisation, resources, supervision, project administration, and writing—original draft, review, and editing.
G López-Castejón: conceptualisation, data curation, supervision, project administration, and writing—original draft, review, and editing.

## Conflict of Interest Statement

The authors declare that they have no conflict of interest.

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
