## [Reviewer comments · Life Science Alliance]

Life Science Alliance

Type I interferon regulates interleukin-1b and IL-18 production and secretion in human macrophages

Rodrigo Diaz Pino, Gillian Rice, Diego San Felipe, Tamar Pepanashvili, Paul Kasher, Tracy Briggs, and Gloria Lopez-Castejon
DOI: <https://doi.org/10.26508/lsa.202302399>

Corresponding author(s): Gloria Lopez-Castejon, The University of Manchester

Review Timeline:

Submission Date:	2023-09-27
Editorial Decision:	2023-10-27
Revision Received:	2024-02-26
Editorial Decision:	2024-03-05
Revision Received:	2024-03-07
Editorial Decision:	2024-03-08
Revision Received:	2024-03-12
Accepted:	2024-03-13

Transaction Report:

October 27, 2023

Re: Life Science Alliance manuscript #LSA-2023-02399

Dr. Gloria Lopez-Castejon
The University of Manchester
The Lydia Becker Institute of Immunology and Infection
Division of Infection, Immunity and Respiratory Medicine
46 Grafton Street
Manchester, Greater manchester M13 9NT
United Kingdom

Dear Dr. Lopez-Castejon,

Thank you for submitting your manuscript entitled "Type I interferon induces expression of inflammasome-related genes in human macrophages but does not impact NLRP3 inflammasome activation" to Life Science Alliance. The manuscript was assessed by expert reviewers, whose comments are appended to this letter. We invite you to submit a revised manuscript addressing the Reviewer comments.

Thank you for this interesting contribution to Life Science Alliance. We are looking forward to receiving your revised manuscript.

Sincerely,

B. MANUSCRIPT ORGANIZATION AND FORMATTING:

Reviewer #1 (Comments to the Authors (Required)):

The study investigates the impact of Type I IFN on inflammasome-related genes as well as on inflammasome activation. This is a relevant topic as very often transcriptional upregulation of inflammasome genes is directly associated with inflammasome activation. However, this is not the case as the activation of the NLRP3 inflammasome requires multiple regulatory steps in addition to the transcriptional upregulation. Data reported in the paper show transcriptional upregulation inflammasome-related genes (including NLRP3, ASC, CASP1, IL1B, IL18 and GSDMD) in interferonopathies, consistent with previous reports. Results obtained in vitro confirmed the transcriptional upregulation and suggest that Type I IFNs do not impact on inflammasome activation.

The work is of interest. However, there are some points that should be addressed:

- previous reports showed that Type I IFNs enhance inflammasome activation in monocytes. In the present paper the authors evaluate the effect on differentiated macrophages. Could it be that the observed differences of the effects of Type I IFNs are due to the different degree of differentiation of macrophages? The authors should discuss this further;
- The fact that LPS treatment reduced the expression of NLRP3 and IL1B in MDMs is unexpected. It is in fact well established that LPS induces upregulation of NLRP3 and IL1b gene transcription. It is possible that the used concentration of LPS is not sufficient to trigger activation of TLR4-dependent transcriptional priming as the authors suggest in the discussion. If this is the case, then I would suggest to confirm key findings of the present work using 1000 ng/ml LPS in order to trigger a more effective priming.
- Overnight stimulation with IFN α significantly increased caspase-1 activation and IL1b release after LPS treatment, suggesting enhanced inflammasome activation. To evaluate whether the NLRP3 inflammasome is involved I would suggest to test whether a selective NLRP3 inhibitor reduced this activation.
- Fig. 5B: how do the authors explain the LDH release after 4h IFN α treatment?
- Fig. 5G. how the authors explain the lack of activation of caspase-1 after LPS+Nig stimulation? This is unexpected;
- Expression of NLRP3 and IL-1b should be also evaluated by WB and Inflammasome activation should also be assessed by evaluating some other endpoints (for example cleavage of caspase-1 by WB and/or ASC speck formation);

Minor:

- In the abstract, ISG should be specified as "interferon-stimulated genes".
- Fig.3C. Is the upregulation of Casp-1 and GSDMD with IFN- α treatment alone statistically significant?
- Fig. 4G Is the increase of cell death and caspase-1 activation, and the increased release of IL1b and IL18 observed after LPS+ATP treatment significant?
- Fig. 5C: is the increase of Caspase-1 activity after LPS+Nig treatment statistically significant?
- Fig. 6F,G: are the differences statistically significant?
- Fig.5A: It is not clear to me why IFN β is indicated in the figure? It seems that the authors only used IFN α in this experiment;
- If I am correct, Fig 4, B-E refers to the same experimental model of Fig3. If this is the case, I suggest not to repeat the scheme of Fig. 4A.

Reviewer #2 (Comments to the Authors (Required)):

The study focuses on investigating the mechanisms of inflammasome activation and regulation by type I interferons (IFNs), which are not well understood. The researchers observed upregulation of inflammasome-related genes, such as Caspase-1 and Gsdmd, and reduced expression of NLRP3 in RNA sequencing data from patients with monogenic interferonopathies. Based on these findings, the study aimed to investigate the effect of type I IFNs on inflammasome regulation in human cells, specifically human monocyte-derived macrophages (MDMs) and THP-1 cells. The major findings of the study are as follows: 1) the observed transcriptional regulation did not translate to the ability of these cells to form an active NLRP3 inflammasome in response to ATP and nigericin, and 2) the data suggests that there is not a strong regulation of the NLRP3 inflammasome by type I IFNs. These findings are relevant to the field of inflammasome research and provide clarity on the role of type I interferons in NLRP3 activation.

I have a few major points that would strengthen the paper:

1. It is known that the priming step of NLRP3 inflammasome activation is associated with increased NLRP3 gene expression and post-translational modifications that control NLRP3 levels and license the NLRP3 protein for inflammasome assembly. While the authors examined NLRP3 gene expression, it would be valuable for the authors to determine the specific modifications exerted on NLRP3 during IFN1a ON vs IFN1ST.
2. In Figure 4C, the authors show increased caspase-1 activation that does not translate to cell death. It would be helpful for the authors to clarify this observation. Western blot analysis showing cleaved Caspase-1 and GSDMD would provide additional support for this conclusion.
3. Supplementary data 1 shows increased Aim2 expression. Could the authors examine the significance of this increased expression by examining the activation of the Aim2 inflammasome with Poly(dA:dT)? This would help clarify the observation.

Reviewer #3 (Comments to the Authors (Required)):

The authors in this study characterize the expression of interferon-stimulated genes and inflammasome components after IFN and/or LPS treatment. The authors observed changes in a set of inflammasome components, most prominently Caspase-1 and NLRP3, depending on the stimulus; however, this did not translate to major changes in inflammasome activation. The studies are thus largely descriptive, and immediate significance wasn't obvious, although these studies could be helpful in dissecting Type I IFN and inflammasome crosstalk between human and mouse macrophages. The experiments are robustly conducted, clear, and logical. Some suggestions below:

1. The introduction is a bit distracting, often delving into too many details of the IFN pathway that are less relevant to this manuscript, thereby diluting the overall message.
2. There is a bit too much attention given to the findings reported by Liu et al., 2017, which weakens the argument for the study. Is there another piece of evidence in the same direction to strengthen the hypothesis and the case to study these pathways?
3. In most of the immunity-related genes, high expression is observed in patient samples. Do the authors have a set of control genes they can test that are not expected to show an increase?
4. In the Abstract: "While the mechanisms of inflammasome activation and type I IFNs are well understood," this sentence is not clear. I believe there are still many unknowns in the field, as evidenced by this study.
5. The last but one sentence in the abstract should be rephrased to say, "Type I IFN does not regulate the NLRP3 inflammasome" or something similar.

We thank all reviewers for their suggestions and comments. Changes in the manuscript as well as responses to reviewers have been included in the text in blue font to allow the reviewers to follow them easier.

Reviewer #1

- Previous reports showed that Type I IFNs enhance inflammasome activation in monocytes. In the present paper the authors evaluate the effect on differentiated macrophages. Could it be that the observed differences of the effects of Type I IFNs are due to the different degree of differentiation of macrophages? The authors should discuss this further.

We also thought that this could be a possibility and have performed the same type of experiments in CD14⁺ monocytes from 7 different healthy donors. We have performed gene expression analysis in response to different IFN- α and IFN- β concentrations as we did for MDMs and observed similar behaviour of gene expression as that observed for MDMs (Fig S2). We have also performed the experiments to determine the level of inflammasome activation in response to LPS and IFN in the absence of a second signal (Fig S4A-D) and found that basal levels of caspase-1 secretion decrease after ON IFN stimulation, both in presence and absence of LPS, unlike what we see in MDMs. We also performed experiments in response to ATP and IFN treatment. We found that caspase-1 activity in the supernatants was significantly decreased after IFN ON treatment in the absence of LPS priming with ATP, and although no significant changes were observed in caspase-1 activity in other conditions, there was a trend to a decrease in caspase-1 activity in the other conditions with IFN- α - ON. We have now included this as supplementary figures Fig S2 and Fig S4, and included in the results section in pages 14, 16, 17 and 18.

- The fact that LPS treatment reduced the expression of NLRP3 and IL1B in MDMs is unexpected. It is in fact well established that LPS induces upregulation of NLRP3 and IL1b gene transcription. It is possible that the used concentration of LPS is not sufficient to **trigger activation of TLR4-dependent transcriptional priming as the authors suggest in the discussion**. If this is the case, then I would suggest to confirm key findings of the present work using 1000 ng/ml LPS in order to trigger a more effective priming.

We found that 10 ng/ml LPS treatment for 6h was sufficient to prime NLRP3 to respond to ATP and nigericin as LPS primed MDMs led to the release of IL-1 β , which did not occur in unprimed cells, to around 3000 pg/ml (Fig 4, 5). We also found that priming with LPS 10 ng/ml potentiated IL-18 release in response to ATP (Fig 4I and 5E, I) compared to unprimed. Other reports have also used 10 ng/ml of LPS to prime their cells and this was sufficient to obtain an appropriate NLRP3 activation with nigericin (10.1136/ANNRHEUMDIS-2020-218366; Supplemental Fig 3) and hence we do not believe that our data is a direct consequence of unprimed or poorly primed cells.

In addition to this we have performed western blot analysis to determine expression of proIL-1 β in MDMs in response to different concentrations of LPS (10, 100 and 1000 ng/ml) for 6h. We found that 10 ng/ml is sufficient to trigger proIL-1 β expression in MDMs and hence should be enough for priming. In some cases, we also observe that

patients already have basal levels of pro-IL1 β but LPS still potentiates its expression. Similar behaviour was found for NLRP3 protein expression. See figure 1 below.

Reviewers comment Figure 1. MDMs were treated for 6h with LPS at indicated concentrations. Levels of NLRP3 and pro-IL-1 β expression was determined by WB with anti-NLRP3 (Cryo-2) anti-IL-1 β antibody (R&D Systems, AF-201-NA). Loading control was determine using anti- α -actin-HRP (Sigma, A3854). Samples from two different donors.

Overnight stimulation with IFN- α significantly increased caspase-1 activation and IL1 β release after LPS treatment, suggesting enhanced inflammasome activation. To evaluate whether the NLRP3 inflammasome is involved I would suggest to test whether a selective NLRP3 inhibitor reduced this activation.

We tried to perform this experiment using NLRP3 inhibitor MCC950 (see below, Fig 2, 3). We tested this experiment after LPS 10 ng/ml or 1 ug/ml priming to see if LPS concentration made a difference in release of caspase1 ad IL-1 mediated by NLRP3. We found that LPS priming concentration did not affect the levels of IL-1 β and IL-18 release. Given the high variability we observe between human samples we have now only detected production of IL-1 β or IL-18 in some of the new samples tested (n=7) probably as we were under the detection limit. In those samples where cytokines were present and potentiated by IFN treatment there is a reduction in IL-1 β and IL-18 release in the presence of MCC950, suggesting it is NLRP3 dependent, however we cannot clearly show that this is the case (Figure 2 reviewers comments). We also tried to detect release of IL-1 β by western blot, but it could not be detected in concentrated supernatants, only in lysates (Fig 3 reviwers comments) and no cleavage could be detected here. Hence, we decided not to include the MCC950 data in the new manuscript version. We have however included the new repeats in the paper (with a n=14 now, Fig 4A-D) to reflect the variability encountered when using the human samples and that this behaviour in response to LPS/IFN might only occur in samples from certain donors.

Reviewers comment figure 2. MDMs from healthy donors were treated with IFN- α (1000 U/ml) overnight (ON, 18 hours) and during the last 6 hours of treatment with type I interferon, were co-treated with LPS (10 ng/ml or 1ug/ml) to induce inflammasome priming and with MCC950 (10uM) to block NLRP3 inflammasome. **(A)** Cell death was measured by LDH assay and shown as fold change relative to UT. **(B)** Caspase-1 release was measured using quantitative assay and shown as fold change relative to UT. **(C)** IL-1 β and **(D)** IL-18 release was measured by ELISA in the supernatants and shown as pg/ml. n =10 independent biological experiments. No statistical differences were observed.

Reviewers comments figure 3. MDMs from healthy donors were treated with IFN- α (1000 U/ml) overnight (ON, 18 hours) and during the last 6 hours of treatment with type I interferon, were co-treated with LPS (10 ng/ml or 1 ug/ml) to induce inflammasome priming and with MCC950 (10 uM) to block NLRP3 inflammasome. IL-1 β expression was measured by Western Blot in cell lysates. β -actin was used as loading control. Showing two different blots from two biologically independent experiments.

- Fig. 5B: how do the authors explain the LDH release after 4h IFN- α treatment?

We cannot really explain why this condition in Fig 5B presents higher LDH release other than increased variability in the samples. However, we did not see any

correlation of this LDH increase with in any of the other outcomes at 4h and hence we do not think it affected any of the conclusions made from this figure.

- Fig. 5G. how the authors explain the lack of activation of caspase-1 after LPS+Nig stimulation?

For Fig 5G we observe a small increase (2-fold) in nigericin treated cells in the absence of IFN indicating inflammasome activation which is supported by induction of IL-1 β and IL-18 release (Fig 5H, I). However, IFN treatment ON induces a higher increase in caspase-1 activity in the supernatants (that we also observe in Fig 4H) that might mask the smaller increase induced by Nig alone.

- Expression of NLRP3 and IL-1b should be also evaluated by WB and Inflammasome activation should also be assessed by evaluating some other endpoints (for example cleavage of caspase-1 by WB and/or ASC speck formation);

We have repeated the experiment to address this comment using samples from 10 additional healthy donors. Unfortunately, variability in human samples meant that detection of caspase-1 and Gasdermin-D was very variable and made it difficult to draw conclusions from these outputs. We have also been unable to detect ASC-specks after ATP activation (5mM, 2h), despite having set up an appropriate immunofluorescence protocol and being able to detect ASC specks in PMA-differentiate THP1 cells after nigericin stimulation (as positive control). Hence, we decided to performed analysis of IL-1 β expression and cleavage in lysates and supernatants by Western blot. We found a lower level of release of mature IL-1 β when MDMs were treated with LPS + ATP in the conditions pre-treated with IFN- α ST or with IFN- α ON compared to no-IFN. We also ran the LDH release, caspase-1 activity as well as IL-1 β and IL-18 release assays in these new set of samples (n=10). After this analysis we found that IL-1 β and IL-18 release measured by ELISA was significantly reduced after IFN treatment, as in the Western blot (while previously there was just a trend). We have now combined this new dataset to the data in our first submission (n=17) to reflect variability of human sample responses. All this new data has now been included in figure 4E-K. We have also amended the text to reflect these changes.

Reviewers comments figure 4. Human monocyte-derived macrophages (MDMs) from healthy donors were treated with IFN- α (1000 U/ml) for 6 hours (ST) or overnight (ON, 18 hours) and during the last 6 hours of treatment with type I interferon LPS (10 ng/ml) was added to induce inflammasome priming and ATP (5 mM) was added in the last 2 hours of the experiment. IL-1 β release and expression was measured by Western blot. β -actin was used as loading control. Showing 2 different biological replicates out of 5 different biological independent experiments

- In the abstract, ISG should be specified as "interferon-stimulated genes".

This has now been updated.

- Fig.3C. Is the upregulation of Casp-1 and GSDMD with IFN- α treatment alone statistically significant? Fig. 4G Is the increase of cell death and caspase-1 activation, and the increased release of IL1b and IL18 observed after LPS+ATP treatment significant? Fig. 5C: is the increase of Caspase-1 activity after LPS+Nig treatment statistically significant? Fig. 6F,G: are the differences statistically significant?

We apologise for not having include these stats. We have now included the statistical differences in all relevant comparison. We have also included this clarification in the methods and figure legends section.

- Fig.5A: It is not clear to me why IFNb is indicated in the figure? It seems that the authors only used IFNa in this experiment;

Yes, only IFN- α was used in the experiment. We have now amended this figure.

- If I am correct, Fig 4, B-E refers to the same experimental model of Fig3. If this is the case, I suggest not to repeat the scheme of Fig. 4A.

Yes, indeed Fig 4 refers to the same experimental model. We have now removed the diagram to amend this figure.

Reviewer #2

1. It is known that the priming step of NLRP3 inflammasome activation is associated with increased NLRP3 gene expression **and post-translational modifications** that control NLRP3 levels and license the NLRP3 protein for inflammasome assembly. While the authors examined NLRP3 gene expression, it would be valuable for the authors **to determine the specific modifications exerted on NLRP3 during IFN1a ON vs IFN1ST.**

As the reviewer suggests it would be interesting to understand the relationship between IFN and NLRP3-regulated control by PTMs. However, as several PTMs (i.e. phosphorylation, ubiquitination, acetylation) have been reported to regulate NLRP3 directly (as well as other components of the inflammasome such as ASC and caspase-1) addressing this would be complicated, and we believe beyond the scope of this particular project.

2. In Figure 4C, the authors show increased caspase-1 activation that does not translate to cell death. It would be helpful for the authors to clarify this observation. Western blot analysis showing cleaved Caspase-1 and GSDMD would provide additional support for this conclusion.

It has been previously reported that IL-1 secretion can occur in the absence of cell death (or low cell death levels). This is the case for alternative NLRP3 inflammasome activity triggered by long LPS priming in human monocytes (10.1016/j.immuni.2016.01.012). Also, IL-1b release in absence of cell death has been reported in DC, in this case mediated by caspase-11 (DOI: 10.1126/science.aaf3036). Activity of caspase-1 in supernatants in those studies was however not measured. It is then possible that caspase-1 activity can be found in supernatants, as is IL-1, in the absence of detectable cell death.

We have tried to detect caspase-1 and Gasdermin D cleaved forms in supernatant and lysates in this experiment, as the reviewer proposed, however we have not been able to detect them possibly due to low level of activation. We also tried to test IL-1 β cleavage in the supernatants. However, although pro-IL-1B was detected in the lysates (see Fig 3 from reviewer's comments), we could not detect IL-1B in the supernatants despite concentrating all supernatant to run these blots.

We have performed this experiment in the presence of MCC950 to answer reviewer's 1 comment (please see above). However, we find very high variability in responses between donors and given the low amounts of IL-1 β /IL-18 detected in some donors in response to IFN it is hard to make clear conclusions on the effect of NLRP3 inhibitor. We still see a group of donors that upregulate caspase-1 and IL-1 in the supernatants after IFN treatment (measured via activity and ELISA) and that MCC950 can reduce this, however this is not significant.

3. Supplementary data 1 shows increased Aim2 expression. Could the authors examine the significance of this increased expression by examining the activation of the Aim2 inflammasome with Poly(dA:dT)? This would help clarify the observation.

As the reviewer indicates, there is increased expression of AIM2 in samples from patients with interferonopathies. It is very plausible that type I IFN is involved in regulation of Aim2 and potentially AIM2 inflammasomes and we agree that this would be an interesting point to study. However, we have observed that the effects of IFN alone in gene expression in our in vitro model, do not match that observed in patients (i.e. NLRP3, IL1B go down) probably due to the complexity encountered in patients with interferonopathies. This paper is focussed on the role of IFN on NLRP3, so we believe that addressing the role of AIM2, although interesting, would be out of the scope of this paper. In addition, dA:dT in human macrophages is able to activate the NLRP3 inflammasome via the cGAS-STING pathway as well as AIM2 inflammasome (10.1016/j.cell.2017.09.039), something we have also reproduced in our lab in THP1 cells in the past. So, these experiments in human cells might not give us the clear experiments and results and that one might need to address AIM2 involvement at the moment.

Reviewer #3

1. The introduction is a bit distracting, often delving into too many details of the IFN pathway that are less relevant to this manuscript, thereby diluting the overall message.

We have now amended the introduction to be more focussed according to the reviewers' comment.

2. There is a bit too much attention given to the findings reported by Liu et al., 2017, which weakens the argument for the study. Is there another piece of evidence in the same direction to strengthen the hypothesis and the case to study these pathways?

Autoimmune disorders associated with elevated levels of IFN such as Systemic Lupus Erythematosus (SLE) <https://doi.org/10.1007/s00251-020-01158-6> and juvenile dermatomyositis (JDM) <https://doi.org/10.1038/s41598-021-04302-8> have been linked to dysregulated inflammasome activation. In addition it has been known for a while that type I IFN is able to suppress expression of IL-1B and IL-1a. It is also known that it can induce the production of the IL-1 receptor antagonist IL-1Ra as well as IL-10, a cytokine known to play an important role in resolving/anti-inflammatory states [10.1038/cmi.2016.25](https://doi.org/10.1038/cmi.2016.25). There are many cases where regulators of NLRP3 inflammasome, some inhibitors for instance, also regulate IL-1b transcription levels (<https://doi.org/10.1124/pharmrev.122.000629>) hence we thought that type I IFN could be one of them and that understanding the crosstalk between IFN and inflammasome will allow to better explain or understand these correlations in disease. When looking at the literature however we could not find many cases in which the study of IFN type I directly on NLRP3 had been approached, and that is why we mainly refer to the Liu and the Guarda studies. However, we agree with the reviewers comment and have now change the text to reflect this (page 5, first paragraph)

3. In most of the immunity-related genes, high expression is observed in patient samples. Do the authors have a set of control genes they can test that are not expected to show an increase?

As interferonopathies are inflammatory conditions many pro-inflammatory genes and ISGs are increased (as expected, PMID: 24183309) in patient samples as shown in Fig 1 and Fig 1S. However, this does not happen to all genes indicating this is not a generalised response. For instance, from the genes shown in Fig S1 IL-1 α and Gasdermin-A gene expression, do not change. In addition to this we have now included a set of 4 genes which expression does not change in these patients (Fig.5 reviewers comments; above) into Fig S1 (GNB1, MARK4, LDHD and C9orf116 (PIERCE1)) to better represent that this is not a generalised event in all genes.

Reviewers comments figure 5: RNAseq was performed from whole blood samples from donors with different mutations, *ACP5* (n=3), *ADAR1* (n=6), *DNASE2* (n=3), *IFIH1* (n=2), *RNASEH2A* (n=3), *RNASEH2B* (n=8), *RNASEH2C* (n=6), *RNASET2* (n=1), *SAMHD1* (n=8), *STAT2* (n=4), *TMEM173* (n=3), *TREX1* (n=8) and *PEPD* (n=3). Expression of *GNB1*, *MARK4*, *LDHD* and *C9orf116* genes was compared between patients with type I interferonopathies and healthy donors and shown as transcripts per million (TPM). control n=5, type I interferonopathies n=42 samples; t-test; non significant for these 4 genes.

4. In the Abstract: "While the mechanisms of inflammasome activation and type I IFNs are well understood," this sentence is not clear. I believe there are still many unknowns in the field, as evidenced by this study.

We have now rephrased this to: "Despite the importance of these cytokines in inflammation, the regulation of inflammasomes by type I IFNs still remains poorly understood."

5. The last but one sentence in the abstract should be rephrased to say, "Type I IFN does not regulate the NLRP3 inflammasome" or something similar.

As the data has slightly changed after new results we have changed the abstract to "We also found that IFN- α treatment reduced the release of mature IL-1 β and IL-18 in response to ATP-mediated NLRP3 inflammasome activation highlighting the importance of tuning IL-1 β and IL-18 secretion levels..."

March 5, 2024

Re: Life Science Alliance manuscript #LSA-2023-02399R

Dr. Gloria Lopez-Castejon
The University of Manchester
The Lydia Becker Institute of Immunology and Infection
Division of Infection, Immunity and Respiratory Medicine
46 Grafton Street
Manchester, Greater manchester M13 9NT
United Kingdom

Dear Dr. Lopez-Castejon,

Thank you for submitting your revised manuscript entitled "Type I interferon regulates inflammasome gene expression and reduces NLRP3-mediated cytokine release" to Life Science Alliance. The manuscript has been seen by an original reviewer whose comments are appended below, and some important issues remain.

Our general policy is that papers are considered through only one revision cycle; however, we are open to one additional short round of revision. Please note that I will expect to make a final decision without additional reviewer input upon re-submission.

Please submit the final revision within one month, along with a letter that includes a point by point response to the remaining reviewer comments.

To upload the revised version of your manuscript, please log in to your account: <https://lsa.msubmit.net/cgi-bin/main.plex>
You will be guided to complete the submission of your revised manuscript and to fill in all necessary information.

B. MANUSCRIPT ORGANIZATION AND FORMATTING:

Sincerely,

Reviewer #1 (Comments to the Authors (Required)):

The authors have performed a high number of experiments in order to address the raised comments. While appreciating the

efforts made by the authors, I think still many critical incongruences remain in the paper.

- Gene expression of NLRP3 and IL1b decreases after LPS priming. I think these data cannot be published as they contrast an established knowledge and mine the concept of priming, which is relevant to this paper. In this respect, the authors found that the protein levels of Pro-IL1b increased after LPS priming and therefore protein expression did not correlate with gene expression. I think there may be some issues with the gene expression experiments. If LPS is not able to induce priming of NLRP3 and IL1b at the gene expression level, I think these experiments cannot be used to establish that "the effect of LPS priming on type I IFN induced gene expression of NLRP3 inflammasome components".

- the authors state that: "IFN can potentiate inflammasome activation in responses to less acute inflammatory signals such as TLR activation alone". The use of MCC950, that I previously suggested, gave highly variable results suggesting that the NLRP3 inflammasome may not be responsible for the observed release of IL1b/IL-18. I think these data are not sufficient to support the conclusions of the authors as the inflammasome may not be involved in the observed release of IL1b. Caspase-1 inhibitor (or the use of ASC ko cells) may help understanding whether other inflammasomes may be involved.

- The inhibition of LPS/ATP-induced IL1b/IL18 release after long term stimulation with IFN is not associated with caspase inhibition. This rules out the possibility that IFN inhibits the NLRP3 inflammasomes, as this inflammasome converges towards caspase-1 activation. Therefore I think that inhibition may occur at cytokine expression level (in fact the author see that IFN inhibits IL1b expression). I think that the title is misleading "Type I interferon regulates inflammasome gene expression and reduces NLRP3-mediated cytokine release".

We have now addressed the comments of this reviewer and included this in the manuscript in purple colour so changes can be easily followed.

The authors have performed a high number of experiments in order to address the raised comments. While appreciating the efforts made by the authors, I think still many critical incongruences remain in the paper.

- Gene expression of NLRP3 and IL1b decreases after LPS priming. I think these data cannot be published as they contrast an established knowledge and mine the concept of priming, which is relevant to this paper. In this respect, the authors found that the protein levels of Pro-IL1b increased after LPS priming and therefore protein expression did not correlate with gene expression. I think there may be some issues with the gene expression experiments. If LPS is not able to induce priming of NLRP3 and IL1b at the gene expression level, I think these experiments cannot be used to establish that "the effect of LPS priming on type I IFN induced gene expression of NLRP3 inflammasome components".

It is well known from the literature that transcript levels by themselves are not always sufficient to predict protein levels (<http://dx.doi.org/10.1016/j.cell.2016.03.014>), meaning that our results do not overly contradict the priming dogma as the reviewer suggests.

IL1B mRNA half-life has been previously determined to be approximately 4h in human PBMC and classical monocytes (CD14+) ([https://doi.org/10.1016/S0021-9258\(18\)86936-8](https://doi.org/10.1016/S0021-9258(18)86936-8); <https://doi.org/10.1038/srep39035>). Moreover it has been shown that despite the reduction in *IL1B* mRNA levels with time IL-1 β protein levels continue to increase over time and hence there is not a direct correlation of *IL1B* mRNA and protein levels ([https://doi.org/10.1016/S0021-9258\(18\)86936-8](https://doi.org/10.1016/S0021-9258(18)86936-8)).

In our study we show that after 6h treatment with 10ng/ml LPS we observe a decrease in *IL1B* mRNA levels, however there is an increase in IL-1 β protein levels at that time point, indicating priming has already occurred. This would fit with the worked described above and suggests that there must be induction of *IL1B* mRNA upon LPS priming. To back this up we have now included our own data showing upregulation of *IL1B* mRNA in MDMs, THP1 and CD14+ monocytes after priming for 4h with 10ng/ml LPS. Here we observe an increase in *IL1B* transcript levels (although not always statistically significant due to human variation), indicating that the transcriptional priming event is happening and can explain the increase in IL-1 β protein expression, and that at 6h a decrease in mRNA levels might have already occurred. These data have now been included as panel B in Fig S3 (see below) and has been included in results (page 16) and discussion (page 21) sections.

Figure Supplementary 3: NLRP3 expression in MDMs and THP-1 cells after treatment with LPS. Human monocyte-derived macrophages, THP-1 cells, and human monocytes CD14⁺ were treated for 4 hours with LPS (10 ng/ml) or left untreated (UT). **(A)** Expression levels of mRNA for *NLRP3* was measured by RT-qPCR and shown as expression relative to UT (fold change). **(B)** Expression levels of mRNA for *IL1B* was measured by RT-qPCR and shown as expression relative to UT (fold change) n=3 biological independent experiments. * = P < 0.05, ** = P < 0.005, *** = P < 0.0005, t-test. *HPRT1* was used as a housekeeping gene. **(C)** MDMs were treated for 6h with LPS at indicated concentrations. Levels of NLRP3 and pro-IL-1β expression was determined by WB with anti-NLRP3 (Cryo-2) anti-IL-1β antibody (R&D Systems, AF-201-NA). Loading control was determine using anti-β-actin-HRP (Sigma, A3854).

- The authors state that: "IFN can potentiate inflammasome activation in responses to less acute inflammatory signals such as TLR activation alone". The use of MCC950, that I previously suggested, gave highly variable results suggesting that the NLRP3 inflammasome may not be responsible for the observed release of IL1b/IL-18. I think these data are not sufficient to support the conclusions of the authors as the inflammasome may not be involved in the observed release of IL1b. A caspase-1 inhibitor (or the use of ASC ko cells) may help understanding whether other inflammasomes may be involved.

We understand the reviewers concerns that we have not shown that this effect is NLRP3 dependent. We agree with this and have indeed not claimed that. We have now amended the discussion to reflect the possibilities that this release might be mediated by NLRP3 or other inflammasomes, or even by other means (Page 21). "However, we still do not know if this effect is induced by activation of NLRP3 as we cannot exclude implication of other inflammasomes such as AIM2 or the involvement of non-canonical inflammasomes". We have also changed the discussion in page 23 to say "... and that IFN can potentiate IL-1 release in responses to less acute inflammatory signals such as TLR activation alone" to reflect this.

We would like to point out that the high variation we observe in these experiments is likely related to the fact that only a group of donors respond to IFN ON by inducing IL-1 release, whilst other donors do not, as reflected by Fig 4A-D. Therefore, no clear conclusions on involvement of NLRP3 could be made, due to intra-donor variation. Therefore, we would expect to see similar variation if repeating this experiment with a caspase-1 inhibitor. Additionally, the use of ASC KO cells would imply using THP1 cells, and these do not reflect the same behaviour as MDMs.

- The inhibition of LPS/ATP-induced IL1b/IL18 release after long term stimulation with IFN is not associated with caspase inhibition. This rules out the possibility that IFN inhibits the NLRP3 inflammasomes, as this inflammasome converges towards caspase-1 activation. Therefore I think that inhibition may occur at cytokine expression level (in fact the author see that IFN inhibits IL1b expression). I think that the title is misleading "Type I interferon regulates inflammasome gene expression and reduces NLRP3-mediated cytokine release".

We understand the reviewer concerns, and we had already added a paragraph in discussion in the previous review (page 22) addressing this *"We observed that IFN- α treatment led to a decrease in IL-1 β and IL-18 levels after NLRP3-inflammasome activation by ATP. However, this pattern was not followed by caspase-1 activity present in the supernatants. This suggests that the decrease in cytokine secretion could be due to the downregulation of the pro-IL1 β and pro-IL18 levels, and not the regulation of the NLRP3 inflammasome"*. We agree with this comment and have amended the abstract and the title to reflect this. The new title is: *"Type I interferon regulates interleukin-1beta and interleukin-18 production and secretion in human macrophages"*

March 8, 2024

RE: Life Science Alliance Manuscript #LSA-2023-02399RR

Dr. Gloria Lopez-Castejon
The University of Manchester
The Lydia Becker Institute of Immunology and Infection
Division of Infection, Immunity and Respiratory Medicine
46 Grafton Street
Manchester, Greater manchester M13 9NT
United Kingdom

Dear Dr. Lopez-Castejon,

Thank you for submitting your revised manuscript entitled "Type I interferon regulates interleukin-1b and IL-18 production and secretion in human macrophages". We would be happy to publish your paper in Life Science Alliance pending final revisions necessary to meet our formatting guidelines.

- please be sure that the authorship listing and order is correct
- please add the Twitter handle of your host institute/organization as well as your own or/and one of the authors in our system
- please consult our manuscript preparation guidelines <https://www.life-science-alliance.org/manuscript-prep> and make sure your manuscript sections are in the correct order
- please move your main, supplementary figure, and table legends after the references section in the main manuscript text
- please add callouts for Figures 4F; S5A-I and S6 to your main manuscript text

A. FINAL FILES:

B. MANUSCRIPT ORGANIZATION AND FORMATTING:

Sincerely,

March 13, 2024

RE: Life Science Alliance Manuscript #LSA-2023-02399RRR

Dr. Gloria Lopez-Castejon
The University of Manchester
The Lydia Becker Institute of Immunology and Infection
Division of Infection, Immunity and Respiratory Medicine
46 Grafton Street
Manchester, Greater manchester M13 9NT
United Kingdom

Dear Dr. Lopez-Castejon,

Thank you for submitting your Research Article entitled "Type I interferon regulates interleukin-1b and IL-18 production and secretion in human macrophages". It is a pleasure to let you know that your manuscript is now accepted for publication in Life Science Alliance. Congratulations on this interesting work.

DISTRIBUTION OF MATERIALS:

Again, congratulations on a very nice paper. I hope you found the review process to be constructive and are pleased with how the manuscript was handled editorially. We look forward to future exciting submissions from your lab.

Sincerely,
